# Osteoclast Fusion: Physiological Regulation of Multinucleation through Heterogeneity—Potential Implications for Drug Sensitivity

**DOI:** 10.3390/ijms21207717

**Published:** 2020-10-19

**Authors:** Kent Søe

**Affiliations:** 1Clinical Cell Biology, Department of Pathology, Odense University Hospital, 5000 Odense C, Denmark; kent.soee@rsyd.dk; Tel.: +45-65-41-31-90; 2Department of Clinical Research, University of Southern Denmark, 5230 Odense M, Denmark; 3Department of Molecular Medicine, University of Southern Denmark, 5230 Odense M, Denmark

**Keywords:** osteoclast, fusion, DC-STAMP, heterogeneity, zoledronic acid, denosumab, CD47, syncytin-1, physiology, pathology

## Abstract

Classically, osteoclast fusion consists of four basic steps: (1) attraction/migration, (2) recognition, (3) cell–cell adhesion, and (4) membrane fusion. In theory, this sounds like a straightforward simple linear process. However, it is not. Osteoclast fusion has to take place in a well-coordinated manner—something that is not simple. In vivo, the complex regulation of osteoclast formation takes place within the bone marrow—in time and space. The present review will focus on considering osteoclast fusion in the context of physiology and pathology. Special attention is given to: (1) regulation of osteoclast fusion in vivo, (2) heterogeneity of osteoclast fusion partners, (3) regulation of multi-nucleation, (4) implications for physiology and pathology, and (5) implications for drug sensitivity and side effects. The review will emphasize that more attention should be given to the human in vivo reality when interpreting the impact of in vitro and animal studies. This should be done in order to improve our understanding of human physiology and pathology, as well as to improve anti-resorptive treatment and reduce side effects.

## 1. Introduction

Osteoclast fusion is a fascinating process that reflects a rare event in the human body—cells merging. Only a few cells are able to merge—at least under physiological conditions. These cells may be myoblasts, trophoblasts, macrophages/giant cells, sperm/eggs, or osteoclasts [1]. Cell fusion is a complex and potentially risky process for several reasons: (1) it is important that only selected cells fuse, (2) that they fuse at the correct site, (3) that they fuse at the right time, (4) that fusion does not result in too large or too small cells, and (5) that fusion is switched off again. If any of these regulatory processes gets out of control, it may have detrimental effects. This is also the case for osteoclasts.

Osteoclasts are the only cells that are able to resorb bone. They do so relatively fast when compared to the much slower reversal/bone formation processes conducted by reversal cells, osteoblasts, and osteocytes [2,3]. Therefore, it is important that osteoclast activity does not get out of control. If this were to happen, there would be a risk of triggering a pathological condition such as osteoporosis. A driving factor for how aggressively osteoclasts can remove a given bone volume is the number of nuclei each osteoclast contains [4,5,6,7,8], and multinucleation is solely determined by osteoclast fusion.

Classically, osteoclast fusion is considered to consist in four basic steps: (1) attraction/migration, (2) recognition, (3) cell–cell adhesion, and (4) membrane fusion. In theory, this sounds like a straightforward simple linear process. However, when considering that this has to take place in a well-coordinated manner—it is no longer simple. In a recent review [9], the complex regulation of osteoclast formation in vivo within the bone marrow was discussed with respect to time and place of differentiation of precursors and their migration to, or arrival at the bone surface where fusion takes place. The present review will focus on osteoclast fusion taking place on/at the bone surface. A special emphasis will be put on placing this in the context of physiology and pathology. In vitro and animal models will be discussed, but will always be considered in the context of human osteoclast biology in vivo. Special attention is given to: (1) regulation of osteoclast fusion in vivo, (2) heterogeneity of osteoclast fusion partners, (3) regulation of multi-nucleation, (4) implications for physiology and pathology, and (5) implications for drug sensitivity and side effects.

## 2. Regulation of Osteoclast Fusion In Vivo

### 2.1. Site-Specific Regulation of Osteoclast Fusion In Vivo

When pre-osteoclasts form in the bone marrow in vivo they can be found in small clusters [10]. These may already be tartrate-resistant acid phosphatase (TRAcP)-positive, but despite a close contact between them, they do not fuse in the bone marrow. A regulatory mechanism must keep them from fusing. Most research efforts to understand fusion mechanisms focus on the differentiation process, more specifically the cytokines and signaling proteins needed for this process get much attention. Such studies have identified major regulators of osteoclast differentiation and fusion (for updated reviews please refer to [11,12])—studies that have revolutionized our understanding of osteoclasts. Yet, it is also interesting to consider why fusion does not take place between two adjacent potential fusion partners. This has received little attention, although it ensures a physiological state of osteoclast fusion; a process that only occurs on the bone surface. 

Under physiological conditions it is central that osteoclasts only form at or on the bone surface determined for resorption [9]. However, in pathologies such as cancer metastases to bone, large osteoclast-like cells form even within the bone marrow without contact to a mineralized bone surface [13,14,15]. In order to understand the mechanisms that prevent this under physiological conditions, it is interesting to consider what characterizes these different conditions. In metastatic cancer such as prostate and breast cancer, cancer cells in the bone marrow produce cytokines such as tumor necrosis factor α (TNFα), interleukin 6 (IL6), and vascular endothelial growth factor (VEGF) that can trigger recruitment and subsequent differentiation. However, it is especially the ability of both cancer forms to trigger the pre-osteoblastic cells to express high levels of receptor activator of nuclear factor kappa-Β ligand (RANKL) and low levels of osteoprotegerin (OPG) that boosts the formation of osteoclasts [16]. In the case of cancer, it is not only the pre-osteoblasts, but also cancer-associated fibroblasts that express RANKL [17,18,19], and these cells are present in large quantities within the cancer-infiltrated bone marrow [20]. This implies that there is a constant and uncontrolled supply of RANKL to the microenvironment. Thus, in these situations there will be uncontrolled osteoclast fusion and subsequently uncontrolled bone loss. Under physiological conditions the location and number of cells expressing RANKL is strictly controlled. In this case, RANKL is expressed by e.g., osteocytes embedded within the bone matrix [21,22], bone lining cells [23], bone remodeling compartment canopy cells [24], and reversal cells [24]. The common denominator of these cells is that they all are located close to, on, or beneath the bone surface. This is one likely way to ensure that osteoclast fusion is only activated at the bone surface. For more information on this issue, please refer to a recent review [9]. Macrophage-colony stimulating factor (M-CSF) is also found to be mostly expressed at the bone surface, but also along vascular structures in the marrow cavity [25,26,27]. This heterogenic distribution of osteoclast precursors exposed to M-CSF and RANKL, with variations in time and space, is likely to create a variety of pre-osteoclasts that express different molecular factors. In addition, they will also display differences in mobility and polarity. 

### 2.2. Heterogeneity Based on Nuclearity, Mobility, Protein Expression, and/or Protein Clusters Regulates and Controls Osteoclast Fusion

As discussed elsewhere [9], the first pre-osteoclast that arrives at the bone surface is likely to get there by migrating through the bone marrow over variable distances. When it reaches the mineralized bone surface, it will switch from being mobile into becoming rather immobile—possibly ensuring that it “waits” for a mobile fusion partner to arrive from the bone marrow. This scenario has been shown to favor fusion [10] and from a physiological perspective it may make sense. In general, heterogeneity between fusion partners with respect to e.g., nuclearity, mobility, expression profile, and/or protein cluster may be a way to regulate and control fusion (Figure 1).

In vitro studies using time-lapse recording investigated the importance of mobility and nuclearity for finding the “right” fusion partner. Using 96 h of time-lapse recording of human pre-osteoclasts and their fusion, it was found [28] that more than 60% of all recorded fusion events occurred between a mobile and an immobile fusion partner. The remaining fusion events occurred between two mobile partners (25%) and two immobile partners (15%), thus, there was a selective bias towards heterogeneity between fusion partners with respect to mobility [28]. Furthermore, another type of fusion partner selection was through multinucleation. In the same study [28], it was found that multinucleation was also a selection parameter. At all stages of differentiation throughout the 96 h there was a bias so that multinucleated osteoclasts in 70% of all fusion events chose a mononucleated fusion partner. Similar findings were also reported in other in vitro studies. Leavot et al. [29] reported, using murine cellular model systems, that so-called fusion-founder pre-osteoclasts will fuse with mononucleated so-called fusion-follower pre-osteoclasts one at a time and in this way reach multinuclearity. Studies focusing on the existence of quiescent pre-osteoclasts in vitro and in vivo suggest that multinucleated osteoclasts only form effeciently if the fusion partner is a mononucleated quiescent pre-osteoclast [30,31]. Further support is obtained from a pioneering study by Jaworski and coworkers [32]. Using a canine (dog) animal model, they demonstrated that existing osteoclasts gained radiolabeled nuclei by fusing with mononucleated pre-osteoclasts one at a time. In a more recent study, Jacome-Galarza et al. [33] reported something very similar, but using a more elaborate technique. They used a mouse model with an inducible deficiency in osteoclast formation (Csfr^fl/−^). Through time–course parabiosis experiments the authors could estimate that osteoclasts primarily became multinucleated by fusing with one pre-osteoclast at a time. Therefore, it seems that there is some consistancy in the in vitro and in vivo findings.

Different levels of multinuclearity are also coupled to the expression of different molecular factors, again creating heterogeneity—a key regulating element of osteoclast fusion. One such example is dendritic cell-specific transmembrane protein (DC-STAMP) [34,35,36,37,38,39]. DC-STAMP is a membrane-bound receptor, but without an identified ligand. DC-STAMP is primarily expressed on the surface of pre-osteoclasts and small osteoclasts [35,36,40]. It has been found that human osteoclast fusion partners display a heterogenic profile for DC-STAMP in the plasma membrane—one fusion partner is positive (mostly mononucleated) and the other negative [40]. This suggests that human osteoclast fusion partners need to be heterogenic for the expression of DC-STAMP in order to fuse. This is supported by studies from mice [34,35]. DC-STAMP-deficient mouse bone marrow-derived pre-osteoclasts demonstrated a complete abrogation of osteoclast fusion [34]. However, DC-STAMP-deficient pre-osteoclasts (DC-STAMP^lo^) were able to fuse with wild-type pre-osteoclasts (DC-STAMP^hi^) [35]. DC-STAMP will be discussed in further detail in Section 4.1. 

Another factor, that is heterogeneously expressed in pre-/osteoclasts and has been shown to be dependent on nuclearity, is CD47. CD47 is, just as DC-STAMP, primarily expressed in pre-osteoclasts and expression decreases as the osteoclast gains nuclei [40,41]. The importance of heterogeneity in CD47 expression between fusion partners was suggested through the work of Hobolt-Pedersen et al. [40], but it was Møller et al. [42] who, through time-lapse, could document that blocking of CD47 selectively inhibited fusion between mononucleated pre-osteoclasts and multinucleated osteoclasts. The opposite result was obtained for syncytin-1, a membrane-bound fusiogen triggering membrane fusion of several human cell types [43,44,45,46,47], including osteoclasts [42,48,49]. The murine variants of syncytin have also been found to trigger membrane fusion of osteoclasts and other cells [50,51,52]. Syncytin-1 in human osteoclasts was found to primarily be involved in fusion between multinucleated osteoclasts [42]; it is thus opposite to DC-STAMP and CD47. Another way of obtaining heterogeneity between fusion partners is the clustered or polarized positioning of factors involved in fusion. A focal up-concentration of both CD47 and syncytin-1 in the plasma membrane has been observed immediately prior to fusion [40]. Such an up-concentration may require membrane structures such as lipid rafts, and for osteoclast this has been shown to involve CD9 [53,54].

When considering all of these, primarily in-vitro-obtained, data, with respect to the importance of heterogeneity (Figure 1), it is important to bear in mind the physiological perspective with respect to regulating osteoclast fusion in vivo. As mentioned, the initial pre-osteoclast arriving on the mineralized bone surface encounters a different microenvironment to that of the pre-osteoclasts that migrate directly from the bone marrow to fuse. This will result in a different expression profile [10], and thus a heterogeneity, which, based on the results discussed above, will disfavor fusion between osteoclasts and pre-osteoclasts on the bone surface, but will instead favor fusion with pre-osteoclasts arriving from the bone marrow. This will also require that the cells relocate their fusion-related factors in a polarized clustered fashion, which may involve factors such as syncytin-1, CD47, and CD9. Thus, as shown in Figure 1, it does seem plausible that a way to ensure a physiological and controlled fusion in vivo, is through the heterogeneity of fusion partners. 

### 2.3. Membrane Fusion during Osteoclastogenesis

It is fascinating to observe cell fusion through e.g., time-lapse recording. It allows you to see how selective the process of finding a potential fusion partner is. It shows how much time it actually takes in vitro for partners to find the “right one”, to orient themselves optimally for fusion, to adhere to the fusion partner, and eventually fuse the two plasma membranes. When it comes to the membrane fusion step itself, one could think that this is a quite straightforward process, but it is not! The hydrophobic cores of the two membranes have to flip and merge in a very hydrophilic environment. In order to circumvent this problem, a number of different approaches to trigger the fusion of two opposing membranes have evolved. These approaches were thoroughly and elegantly presented in a recent review [12] and will, in the present review, therefore not be elaborated in more detail. Regarding more specific details on membrane fusion mechanisms and fusiogens for multiple cell types, reviews by Podbilewicz and co-authors can be recommended [1,55]

## 3. Osteoclast Fusion In Vivo Using Animal Models

Through the use of intravital two-photon microscopy in mice, Nevius and colleagues [56] observed that in vivo fusion occurs between a larger rather immobile osteoclast and a smaller mobile pre-osteoclast. These smaller mobile pre-osteoclasts were found to migrate from the bone marrow towards the bone surface, fusing with already-existing osteoclasts. They proposed that the attraction of these mobile pre-osteoclasts is caused by secretion of 25-hydroxycholesterol by osteoblast lineage cells situated on the bone surface. 25-hydroxycholesterol acts as a ligand for the Gαi protein-coupled receptor, EBI2 [56]. Based on abstracts presented at international conferences it seems that there is some further progress on its way. Dallas and colleagues [57] used ex vivo imaging of calvarial explants from tdTomato LysM-Cre-expressing mice and observed very dynamic osteoclast behavior. They reported that fusion between adjacent osteoclasts could be observed, apparently on the bone surface. McDonald and colleagues also reported similar findings. Using intravital tibia multiphoton microscopy they observed [58] fusion between adjacent osteoclasts, again apparently on the bone surface. This was especially prominent when animals were given an injection of RANKL—so apparently, it must have been less obvious under physiological conditions. Thus, these very new findings may suggest that in mice, osteoclasts may choose to fuse with adjacent osteoclasts on the bone surface. However, this was not explicitly mentioned by Nevius et al. [56], who only reported that pre-osteoclasts migrating from the bone marrow were observed to fuse with existing osteoclasts on the bone surface. Maeda et al. [59] also reported, using intravital imaging, that under physiological conditions, murine calvarial osteoclasts were individual and rather small osteoclasts, but if mice were injected with RANKL they rapidly became far larger. The study of Jacome-Galarza et al. [33] indirectly supported that fairly large osteoclast syncytia seem to exist on the bone surface of adult mice. However, they explicitly concluded that these osteoclasts gain nuclei through fusion with single pre-osteoclasts from the marrow/circulation. Yet, since the studies of McDonald et al. [58] and Maeda et al. [59] found that injection of exogenous RANKL triggered existing osteoclasts to become abnormally large, it is possible that a key to regulating osteoclast multinuclearity is through a very strict regulation of RANKL expression. It could be speculated that existing osteoclasts still need an exposure to RANKL in order to continue to fuse, but if there is no RANKL, there is no fusion, or at least, very slow fusion. Furthermore, some of these new in vivo studies also bring in fission and subsequent “refusion” as an element to regulate nuclearity in vivo [57,58], thereby putting preceding in vitro findings into an in vivo perspective [60,61]. Given the novelty of many of these in vivo observations and the limited number of published studies, more research is needed in order to fully understand the implications of the recent findings using intravital microscopy. In this regard, it is always desirable to evaluate the relevance of such astonishing findings from animal studies to the human context. It will be interesting to follow the progress in this regard in the coming years. 

### Longevity of Mouse Osteoclasts In Vivo

It is possible that the life span of osteoclasts in vivo, in particular in mice, will be longer than previously considered. Mouse osteoclasts in bone explant cultures or in vitro cultures in general live for less than one week, but life spans in vivo may be different—considerably different. Through the very impressive work of Jacome-Galarza and co-authors [33], they were able to obtain data supporting that osteoclasts may stay alive and be functional for multiple months. This knowledge was obtained through e.g., the injection of EdU^+^ monocytes, tracing their incorporation into osteoclasts, and determining for how long they could be detected in osteoclasts. Much earlier studies performed irradiation of the animal to eliminate osteoclast precursors in the marrow and traced for how long preexisting osteoclasts could be observed. Based on this approach they estimated that in mice the life span of osteoclasts was less than six weeks. However, if they were to continue to fuse, their lifespan may be longer [62]. Using pulse labelling of post-mitotic nuclei with ^3^HTdR in dogs, it was found that existing osteoclasts in cortical bone may live for at least two weeks, but a maximum life span was not determined [32]. Also, pioneering work [63] supported that osteoclasts in mice are long-lived and that their life can be prolonged by regular fusion with mononucleated pre-osteoclasts. So, based on both present and past research it seems reasonable to conclude that individual osteoclasts in various animal models may stay alive for multiple weeks and months and that they can do so through occasional fusion with pre-osteoclasts, thereby refreshing their nuclei pool. However, do they become more multinucleated in this process or do they discard individual nuclei through controlled apoptosis of individual nuclei? Jacome-Galarza and co-authors [33] did see a modest increase in the number of nuclei per osteoclast within a five-month period. Could this suggest that osteoclasts with a longer life span would gain nuclei and thereby become very large? Our knowledge in this regard seems limited, but future studies will hopefully shed more light on these new and very interesting observations.

When it comes to longevity of human osteoclasts, we basically have no knowledge, but it is classically stated that once an osteoclast has finished a resorption cycle it will undergo apoptosis and eliminate itself, thus there is an expected lifespan of a few weeks [64,65,66,67,68]. Available evidence from adult human bone points to osteoclasts existing as individual cells of limited size. Through 3D reconstruction of serial histological sections stained for TRAcP-positive cells on the bone surface, it was clearly observed that osteoclasts exist as individual osteoclasts of limited size on the bone surface of both cancellous [69,70,71] and cortical bone surfaces [72]. These were analyzed in bones from patients with different pathologies such as multiple myeloma [70] and osteoporosis [71], but also from healthy controls [10,69,71,72]. Their limited size in human adults under physiological conditions could be interpreted as osteoclasts being short-lived in vivo, because longevity of osteoclasts is favored by continuous fusion [33,62,63]. Thus, it seems straightforward to conclude that the human osteoclast life span in vivo is rather short and is regulated through apoptosis [64,65,66,67,68]. However, strictly speaking, we cannot know for sure, simply because no one has yet specifically looked for it. It is possible that osteoclasts will not grow in size, although they will regularly fuse to enable longevity if selected nuclei are simultaneously discarded one by one. Therefore, more research and alternative approaches may be necessary to transfer the fascinating mouse model data into a human setting.

## 4. Regulation of Multinucleation

It is known that osteoclasts with many nuclei are more aggressive [4,5,6,7,8]. It is therefore very important to understand how continued and uncontrolled osteoclast fusion is avoided. However, very little research has focused on how fusion stops again—most research efforts are invested in understanding how fusion is mediated. 

In order to understand this regulation, it may be helpful to consider the pathology of Paget’s disease, where osteoclasts fuse uncontrolled [73,74]. Although the mechanism causing this uncontrolled fusion is not fully understood, genetic studies have found several genes involved. The most common mutation occurs in the sequestosome 1 (*SQSTM1*) gene leading to an inactivation of the protein [74]. An inactivation of SQSTM1 can result in an amplification of e.g., RANKL signaling because SQSTM1 is involved in preventing the activation of NF-κB-mediated signaling [74,75]. This suggests that a main negative regulator of osteoclast multinucleation may simply be through the regulation of signaling. This is not surprising, but classically such regulators would be identified intracellularly. However, one may also consider the spatial and temporal expression of cytokines such as M-CSF and RANKL, but also a negative regulator such as OPG, in vivo. A spatio-temporal control may be an effective way to negatively regulate fusion, preventing osteoclasts from getting too large. This precise issue was discussed in a recent review [9] and will therefore not be elaborated more here, but recent developments have brought forward new angles on the regulation of osteoclast multinucleation in vivo, something which will be discussed below.

### 4.1. DC-STAMP—A Master Regulator of Osteoclast Multinuclearity

Other genes than *SQSTM1* were also found in genome-wide association studies (GWAS) to correlate with Paget’s disease. Several of these genes encode receptors or ligands known to play a role in osteoclast fusion—such as M-CSF, RANKL, and DC-STAMP [76,77,78]. DC-STAMP is known as a master regulator of osteoclast differentiation and fusion. An overview of its molecular actions was recently presented in detail [11]. Much of this knowledge comes from studies in cell culture and animal models. Such studies give a good understanding of how DC-STAMP mediates its regulating effects. However, they do not say much about how DC-STAMP may be involved in physiological and pathological conditions, but there are some exceptions. The finding that single nucleotide polymorphisms (SNPs) in the *DCSTAMP* gene correlates with Paget’s disease, suggests that this gene may not only be responsible for facilitating osteoclastogenesis, but that it also represents a “tool” that the cell/body can use to regulate when a desired osteoclast nucleation level is reached. Given that Paget’s disease is a condition where multinucleation of osteoclasts is out of control, it is interesting that precisely the *DCSTAMP* gene contains a SNP (rs2458413) that can trigger this condition [77,78]. The fact that this SNP (rs2458413) causes multinucleation suggest that it either results in a gain of function or an overexpression. Indeed, Mullin and co-workers [78] found that the SNP rs2458413 resulted in a 50% upregulated gene expression, supporting that it may be relevant to look at the expression level of DC-STAMP as an intrinsic regulator of osteoclast nucleation in vivo. Another SNP in the *DCSTAMP* gene, rs62620995, was also found to be linked with Paget’s disease [79] and osteoclasts generated from carriers of this SNP specifically gained more nuclei per osteoclast than non-carriers [80]. A recent study [7] used blood from 49 women to generate osteoclasts in vitro. It was found that the gene expression levels of *DCSTAMP* of in-vitro-generated osteoclasts correlated positively with the in vivo C-terminal telopeptide of type I collagen (CTX) levels of the same donors. Furthermore, *DCSTAMP* gene expression also correlated positively with the nucleation level of the osteoclasts in vitro, the nucleation level in vitro correlated with CTX levels in vivo, and last, but not least, DC-STAMP gene expression in vitro correlated positively with the age of the donors (44 to 66 years) [4,7]. This strongly suggests that DC-STAMP may be a cornerstone in understanding why bone resorption increases with age. How may this be regulated in vivo? Data suggest that this may be through epigenetic regulation [4,7,81,82,83,84,85,86]. An epigenetic regulation of the *DCSTAMP* gene is interesting, because it may be a tool to regulate this powerful regulator of osteoclast multinucleation both through DNA methylation [7,81], histone methylation [86], and miRNA [82,83,84,85]. Studies of Møller et al. [4,7] have shown that DNA methylation levels of the *DCSTAMP* gene are lower with increasing age and that this results in an elevated gene expression, multinucleation, and bone resorptive activity of osteoclasts in vitro. However, since this activity is also linked to the in vivo CTX-levels of the pre-osteoclast donor, this strongly suggest that the in-vitro-obtained data reflect a relevant physiological finding [7].

### 4.2. OSTM1—A Negative Regulator of Osteoclast Multinuclearity

In a recent review [87], it was highlighted how studies over the years have identified osteopetrosis-associated transmembrane protein 1 (Ostm1) as a regulator of multinucleation. The osteopetrotic phenotype of the grey-lethal (*gl*) mouse as well as a variant of severe autosomal recessive infantile malignant osteopetrosis were both caused by a gene defect in the mouse *Ostm1* and human *OSTM1* gene [88] resulting in a partial deletion leading to a null allele. It is interesting that this gene defect results in large but inactive multinucleated osteoclasts both in vitro and in vivo. Thus, Ostm1 is a negative regulator of osteoclast multinucleation. It was found to be a negative regulator of the Nfatc1 pathway and thereby also a negative regulator of *DCSTAMP* gene expression [89,90]. The fact that large multinucleated osteoclasts, in the case of *OSTM1* mutations, are also non-functional further highlights that a proper regulation of multinucleation is key. Ostm1 is also linked to the acidification function of osteoclasts—a central element allowing the osteoclast to degrade bone since low pH is essential for the osteoclast to dissolve mineral and degrade collagen efficiently [91,92]. Proper acidification is linked to the coordinated activities of the chloride pump, CLC7, and the proton pump, H^+^vATPase [93,94]. Ostm1 forms a complex with CLC7 and is essential to acidify lysomomal vesicles in osteoclasts [93,94,95,96]. Although deficits in CLC7 function have not been related to osteoclast fusion, the activity of vATPase has [97]. However, the actions of Ostm1 and H^+^vATPase have opposite actions on fusion so this adds a level of complexity to understanding how lysosomal involvement may play a role in multinucleation.

## 5. Implications for Physiology and Pathology

### 5.1. Multinucleation of Osteoclasts in Physiology

What are the direct consequences of osteoclast multinucleation with respect to physiology? Certainly, a multitude of animal studies have investigated the consequences of presence or absence of a long list of molecular factors that are key for osteoclast multinucleation. Yet, despite the many impressive discoveries made with this type of investigation, we must also acknowledge that removing a factor completely may throw off the fine-tuned balance between networks regulating osteoclast multinucleation. This may then cause other factors to be altered. Therefore, in order to learn about the importance of various molecular factors for osteoclast multinucleation under physiological conditions, an alternative way may be to use the natural variation between not only humans, but also rodents. This may be an alternative approach that allows us to interpret the importance of molecular factors of osteoclast multinucleation in real life. GWAS analyses in both human and animal studies could be such an approach.

Good examples of such an approach used the genetic variation between two strains of rats with a different bone phenotype, and tested the potency of osteoclastogenesis in vitro from bone marrow progenitors [8,98]. Through expression quantitative trait loci (eQTL) analyses researchers found a trans-regulated gene network consisting of 190 osteoclast-expressed genes regulated by a master-regulator, triggering receptor expressed on myeloid cells 2 (Trem2) [98]. This clearly shows the complexity of osteoclast multinucleation mechanisms. Using the two different rat strains, Wistar-Kyoto and Lewis, it was found that an altered expression of e.g., *Dcstamp* and *Cd9* were one of the causes of the different bone phenotypes in these two rat strains [98]. Thus, these findings link to the discussions in Section 2.2 and Section 4.1. In a later study, the same study group used the network of 190 osteoclast genes identified in rats to compare with GWAS findings in a human study cohort [8]. It was found that SNPs from this cluster were enriched especially for the traits of heel bone mineral density and body height. One of the very prominent genes was *DCSTAMP*. Thus, genetic variations in this gene both in rat and man are at least in part responsible for bone density and growth under physiological conditions.

### 5.2. Multinucleation of Osteoclasts in Pathology and Anti-Resorptive Treatment

Bisphosphonates are long-standing drugs used to target osteoclasts in order to stop their bone destructive activity under pathological conditions. These may be e.g., osteoporosis, cancer metastases to bone, and multiple myeloma—just to mention a few. The mode of action of bisphosphonates is linked to the mevalonate pathway. Inhibition of this pathway will prevent prenylation of membrane-anchored signaling proteins and cause an accumulation of pyrophosphate. Both of these outcomes have been described to inhibit bone resorption activity and/or trigger apoptosis [99]. The mevalonate pathway is a key pathway for osteoclast function (as well as other cell types) and a blockage of this pathway should be able to overpower potential compensating effects. Yet, it may not be as simple as that. In a recent in vitro study [100] using osteoclasts generated from 46 female blood donors, a >210-fold variation was reported in the osteoclasts’ sensitivity (from these different individuals) to zoledronic acid in vitro. In a multiple linear regression analysis including ten independent variables, the multinucleation level of osteoclasts in vitro was found to reduce sensitivity. For each nucleus gained, the IC50 increased. This is intriguing and has (as far as the author knows) not been observed before. However, this may suggest that the regulation of osteoclast multinucleation in vivo may be a parameter to consider, with respect to drug resistance to osteoclast-targeting treatments in vivo.

Studies that show the efficiency of zoledronic acid or alendronate treatment of patients in general show that these are very potent drugs. However, they are not equally potent [101,102,103,104,105]. One possible interpretation could be that there may be resistant pools of osteoclasts that, at least in part, escape full inhibition. Bisphosphonates in general are found to reduce the number of osteoclasts considerably, but the cells that do remain have, both in animal models [106] and humans [107,108,109,110], been reported to be more rich in nuclei than without treatment. Underneath these osteoclasts, signs of a diminished, yet still present, bone resorptive activity was detected [107,108], and patients with many of these large osteoclasts showed more residual bone resorption than those where these osteoclasts were not observed [110]. This could suggest that large giant multinucleated osteoclasts are able to retain some residual bone resorptive activity as an indication of partly-reduced sensitivity. Large multinucleated osteoclasts are also often seen in the context of cancer bone metastases [111,112]. Precisely in cancer, suppression of bone resorption is important to treat the bone disease. Yet, although bisphosphonates are indeed effective for treating bone resorption activities in many patients, they are not equally potent in all [103,105,113] and this may have detrimental effects for these patients. GWAS investigations have been undertaken to look for genetic predisposition to an altered sensitivity to bisphosphonate treatment. However, these studies have primarily searched for a genetic predisposition to side effects such as atypical femur fractures or osteonecrosis of the jaw [114,115,116,117]. One of these studies is particularly interesting because it identified SNPs in the all-trans retinoic acid induced differentiation factor (*ATRAID*) gene to potentially predispose to both atypical femur fractures and osteonecrosis of the jaw [115]. This is interesting because *ATRAID* was reported to encode for a transporter in the lysosomal membrane that is responsible for releasing bisphosphonates into the cytoplasm of the cell, where they can act on the mevalonate pathway [118]. This makes a direct link to the action of bisphosphonates, but because the SNP is so rare [115,118], it is unlikely to account for the widespread variations in sensitivity to bisphophonates in vivo [103,105,113] and osteoclasts in vitro [100]. Rather it may be of relevance to use an epigenome-wide association study (EWAS) approach to look for explanatory and predictive profiles for bisphosphonate sensitivity. As discussed in Section 4.1, Møller and colleagues reported [4,7] that the *DCSTAMP* gene is epigenetically regulated (DNA methylation) during ageing, resulting an increased gene expression and an increased nucleation of osteoclasts in vitro. This phenomenon was positively correlated with bone resorption levels in vivo. As aforementioned, the same group showed that more-multinucleated osteoclasts are more resistant to zoledronic acid than those with less nuclei [100]. Thus, these studies suggest a possible link between epigenetic control and sensitivity to zoledronic acid. It would be interesting to consider this possibility in future EWAS studies, something that apparently has not yet been done. 

Recently, denosumab has received quite some attention due to general concerns about the so-called “rebound effect” observed after discontinuing treatment with denosumab. Especially after prolonged treatment [119,120], there are clinical concerns that treatment with the most potent bisphosphonate, zoledronic acid, is not sufficient to prevent the rapid bone loss observed after discontinuation of denosumab treatment [121,122]. In animal models of denosumab treatment, a rebound effect was observed after stopping OPG treatment resulting in the reformation of numerous osteoclasts [58], something which is supported by numerous reports on discontinuing denosumab treatment for patients [119,120,123]. Up to now, no reasonable explanation has been given as to why zoledronic acid is not able to prevent this bone loss, but only to dampen it. In conjunction with the discussion on zoledronic acid efficiency, it would be interesting to consider if the findings presented by Møller et al. [100] may suggest a mechanistic explanation. 

## 6. Conclusions

When trying to understand osteoclast fusion we have come far by identifying central cytokines and signaling pathways that are essential for the generation of osteoclasts both in vitro and in vivo. However, more attention should be given to the human in vivo reality when interpreting the impact of in vitro and animal studies. This should be done in order to improve our understanding of human physiology and pathology, as well as to improve anti-resorptive treatment and reduce side effects. This is precisely the aim of the present review. This review also wishes to emphasize that with a focused and passionate engagement into understanding human osteoclast dynamics in vivo, we are not far from generating tools that will allow us to individualize treatment with anti-resorptive drugs, to the benefit of patients. While we pursue our quest to find new drug targets and new drugs, it is also worthwhile to invest in optimizing and individualizing the use of the drugs we already have.

## Figures and Tables

**Figure 1 ijms-21-07717-f001:**
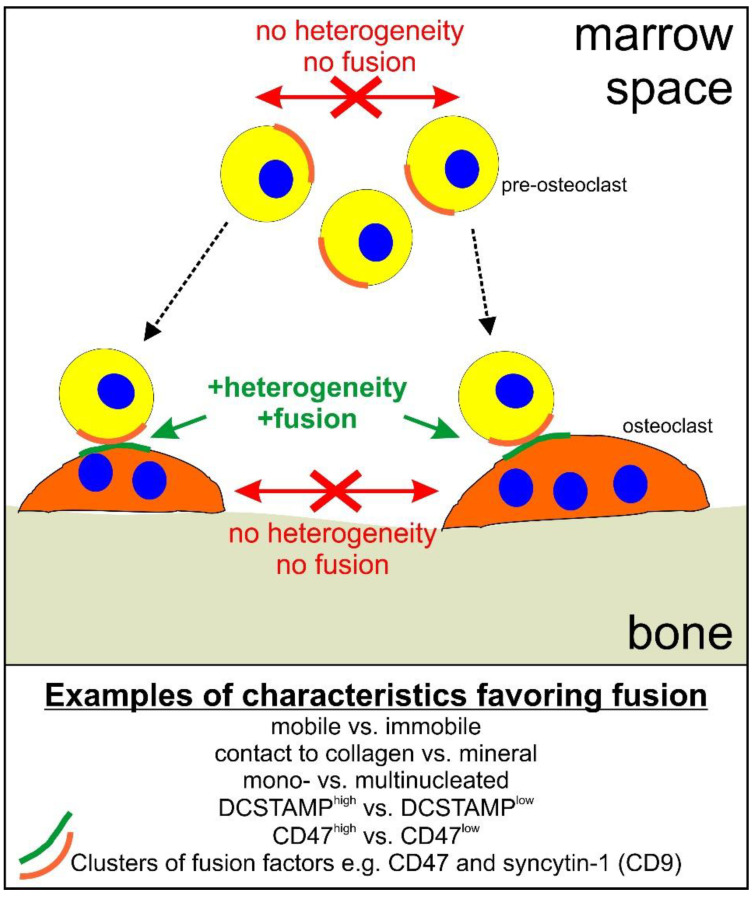
Schematic representation of how heterogeneity may regulate osteoclast fusion. Dotted arrows indicate migration of pre-osteoclasts to the bone surface. Green text and arrows indicate that heterogeneity favors fusion. Red arrows, crosses, and text indicate that lack of heterogeneity reduces the likelihood of fusion.

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
