# Peer review of "Osteoclast Fusion: Physiological Regulation of Multinucleation through Heterogeneity—Potential Implications for Drug Sensitivity"

_ijms, 2020, doi:10.3390/ijms21207717_

Round 1

Reviewer 1 Report

This review article by Kent Søe is generally well written, clear, and although very focused, it is potentially of interest for the field.

Despite being interesting, there are some aspects that need to be improved before publication, as listed below:

  • Page 1 line 32: a. b. c. should be changed with something which more clearly suggests this is a list of items, e.g. i) ii) iii) or 1) 2) 3)
  • Page 1 line 34: change “either” with “any”
  • Page 1 line 35: change “also in the case of osteoclasts” with “And this is also the case for osteoblasts”
  • Page 2 line 50: change “References to in vitro studies and animal models will be plenty” with “In vitro and animal studies will also be discussed”
  • Page 2 line 64: the capitalized “not” is a bit too flamboyant and may drive people to look it up in the abbreviations list
  • Page 2 line 68: I would comment that these cells cannot be called osteoclasts, but osteoclast-like cells, since a bone fide osteoclast should actually be located on the bone surface. I would include this change throughout the manuscript
  • Page 3 line 96: change “into” with “to”
  • Page 3 line 94-100: although it presents concrete data, this section looks too speculative – there is data showing the migration patterns described, and they should be referenced (2 references for all these mechanisms is just too few), otherwise this section should be simplified and perhaps incorporated in other paragraphs. Also, perhaps a more befitting title for this paragraph would be “behavioural heterogeneity of pre-osteoclasts and/or osteoclasts as a tool to control fusion” otherwise one could think you are talking about different cells, whereas these are the same cells behaving differently
  • The words in vitro and in vivo should be italicized
  • When quoting somebody’s work by name, be consistent. Either name and surname, or just surname
  • Page 5 line 214: typo “Longevity”
  • Page 5 line 215: change “will be longer than originally anticipated” with “is longer than previously hypothesised” and quote at least one or two references referring to osteoclast longevity
  • Page 8 line 352: interesting observation, might that just be due to the fact that the ratio between cytoplasm and membrane increases, lowering intracellular bisphosphonates concentration due to slower drug exchange across the membrane?
  • Perhaps a paragraph exploring the possibility of a targeted anti-fusion therapy associated with a classical anti-resorptive therapy could give this review more breadth and scope.
  • Page 9 lines 415-418: perhaps you should stop at the first “osteoclasts”…you can say the rest in person, but I do not think it is appropriate for publication
  • An introductory paragraph about bone biology should be included
  • How does the bone marrow microenvironment influence osteoclast fusion, and how is this relevant to human disease? The contribution of at least immune cells and osteoblasts should be discussed in this review
  • The concept of osteoclast fission should be discussed in further detail, since it has been gaining lots of attention lately
  • I suggest including at least one (or more) summary cartoon(s)/graphical abstract: it would greatly help the reader follow the manuscript and capture its meaning

Reviewer 2 Report

     This review summarized the current research of osteoclast fusion at the physiological state in animals and humans and tried to understand the reality of  osteoclast fusion in various human bone diseases. It is well organized according to the spatio-temporal sequence of osteoclast fusion. The cited papers are suitable and comprehensive. 

Major comments:

  1. Page 5, section 3.1, Longevity of osteoclasts in vivo of mice, Second paragraph.

     The author used the term “separated non-syncytia like osteoclastss” (line 232), “as isolated osteoclasts” (line 233), and “human osteoclast syncytia” (line 239) for the first time.  I think osteoclasts are syncytia. What do you mean by using “non-syncytia like osteoclasts”? In addition, “separated or isolated osteoclasts” would confuse readers. Do you want to say “The number of osteoclasts in the reverse-resorption phase are small compared to that in the initial resorption phase” (Ref. 3)? Furthermore, the cited papers did not mention the life span of human osteoclasts (Ref. 63-66).  Why do you think that human osteoclasts have the short life span (line 238)? It would be nice to speculate a difference in the life span of human osteoclasts between the initial and secondary resorption phase. Do the osteoclasts generated in the first resorption phase transmigrate to the secondary resorption phase? The author is encouraged to solve these concerns. 

Minor comments:

  1. Page 3, line 106.   “favors fusions” is “favors fusion”?
  2. Page 4, line 153.  “fusions” is “fusion”?
  3. Page 4, line 191.  “fusions” is “fusion”?
  4. Page 5, line 214.  “Longgevity” is “Longevity”.
  5. Page 5, line 242.  “multi-nucleation” is “multinucleation”.
  6. Page 6, line 272.  “a desired nucleation of level of  osteoclasts” is “a desired level of osteoclast nucleation”?
  7. Page 7, line 295.  “the pre-osteoclast donor this strongly suggest that” is “the pre-osteoclast donor, this strongly suggests that”.
